# Rapid and Efficient Enrichment of Snake Venoms from Human Plasma Using a Strong Cation Exchange Tip Column to Improve Snakebite Diagnosis

**DOI:** 10.3390/toxins13020140

**Published:** 2021-02-13

**Authors:** Chien-Chun Liu, Ya-Han Yang, Yung-Chin Hsiao, Po-Jung Wang, Jo-Chuan Liu, Chien-Hsin Liu, Wen-Chin Hsieh, Chih-Chuan Lin, Jau-Song Yu

**Affiliations:** 1Molecular Medicine Research Center, Chang Gung University, Taoyuan 333, Taiwan; chienchunliu1016@gmail.com (C.-C.L.); hschin@mail.cgu.edu.tw (Y.-C.H.); mapleleaves730@gmail.com (P.-J.W.); 2School of Medicine, College of Medicine, Chang Gung University, Taoyuan 333, Taiwan; yangyahan8246@gmail.com; 3Liver Research Center, Chang Gung Memorial Hospital at Linkou, Taoyuan 333, Taiwan; 4Graduate Institute of Biomedical Sciences, College of Medicine, Chang Gung University, Taoyuan 333, Taiwan; Joy6408@gmail.com; 5Center for Research, Diagnostics and Vaccine Development of Centers for Disease Control, Ministry of Health and Welfare, Taipei 10050, Taiwan; liuch@cdc.gov.tw (C.-H.L.); vac@cdc.gov.tw (W.-C.H.); 6Department of Emergency Medicine, Chang Gung Memorial Hospital at Linkou, Taoyuan 333, Taiwan; bearuncle@yahoo.com; 7Research Center for Food and Cosmetic Safety, College of Human Ecology, Chang Gung University of Science and Technology, Taoyuan 333, Taiwan

**Keywords:** snakebite, venom, strong cation exchange chromatography, lateral flow strip, clinical diagnostic assay

## Abstract

Snake envenomation is a serious public health issue in many tropical and subtropical countries. Accurate diagnosis and immediate antivenom treatment are critical for effective management. However, the venom concentration in the victims’ plasma is usually low, representing one of the bottlenecks in developing clinically applicable assays for venom detection and snakebite diagnosis. In this study, we attempted to develop a simple method for rapid enrichment of venom proteins from human plasma to facilitate detection. Our experiments showed that several major protein components of both *Naja atra* (*N. atra*) and *Bungarus multicinctus* (*B. multicinctus*) venoms have higher isoelectric point (pI) values relative to high-abundance human plasma proteins and could be separated via strong cation exchange–high-performance liquid chromatography (SCX-HPLC). Based on this principle, we developed an SCX tip column-based protocol for rapid enrichment of *N. atra* and *B. multicinctus* venom proteins from human plasma. Application of liquid chromatography-tandem mass spectrometry (LC-MS/MS) led to the identification of cytotoxin and beta-bungarotoxin as the major proteins enriched by the SCX tip column in each venom sample. The entire process of venom enrichment could be completed within 10–15 min. Combination of this method with our previously developed lateral flow strip assays (rapid test) significantly enhanced the sensitivity of the rapid test, mainly via depletion of the plasma protein background, as well as increase in venom protein concentration. Notably, the SCX tip column-based enrichment method has the potential to efficiently enrich other *Elapidae* snake venoms containing proteins with higher pI values, thereby facilitating venom detection with other assays. This simple and rapid sample preparation method should aid in improving the clinical utility of diagnostic assays for snakebite.

## 1. Introduction

Snake envenomation is a serious public health issue in many tropical and subtropical countries. According to a recent epidemic study, about 1.8–2.7 million snake envenomation cases occur each year, leading to 81,410–137,880 deaths and 244,230–413,640 amputations or other permanent disabilities [1]. In Taiwan, approximately 1000 snake envenomation cases are reported on an annual basis. Six species of clinically important venomous snakes are found in Taiwan, which include *Trimeresurus stejnegeri, Protobothrops mucrosquamatus, Naja atra, Bungarus multicinctus, Deinagkistrodon acutus,* and *Daboia russelli formosensis* [2,3]. Four types of antivenom have been produced to treat snakebites by these native species [4], and, due to the sufficient supply, antivenom is always given for envenoming in Taiwan.

Several clinical challenges need to be overcome for effective management of snakebite. Firstly, victims bitten by different snakes may display similar symptoms, leading to potential misuse of antivenom treatment. Snake venom can cause irreversible damage if patients do not receive the correct treatment [5], highlighting the vital need for accurate diagnosis and immediate intervention. Currently, two main techniques are available for the detection and diagnosis of snake envenomation. The first is enzyme-linked immunosorbent assay (ELISA) with high sensitivity that has been successfully used for clinical diagnosis of snakebite and detection of venom concentrations in patient plasma [6,7]. ELISA is highly sensitive, with a detection limit of up to 1 ng/mL. However, this technique takes more than four hours to complete, which is too time-consuming for snakebites that require immediate treatment. Another common method is the rapid test [8,9,10], which usually takes 5 to 20 min to generate results, a time-frame more suitable for effective snakebite diagnosis and management. However, this procedure is less sensitive than ELISA, with a detection limit of ~5–10 ng/mL. Even in cases where the venom level in serum/plasma is lower than 10 ng/mL, victims may develop local and/or systemic symptoms. Moreover, since the observed serum/plasma venom protein concentration decreases with time, prediction of clinical severity according to this parameter is a challenge [7]. Difficulties in detection might occur if confronting patients that delayed in seeking medical care or patients with low venom concentration in their plasma. Therefore, the sensitivity of the rapid test should be enhanced in order to improve its clinical utility.

In this study, an SCX tip column-based method was developed to rapidly enrich snake venom proteins from human plasma based on biophysical differences (pI values) between human plasma proteins and main venom proteins. Combination of this method with rapid test increased the sensitivity of the rapid test and enriched signals to facilitate identification of the envenoming species. We propose that the protein enrichment technique developed in this study may be successfully applied to enhance the sensitivity and clinical utility of biological assays for snake venom.

## 2. Results

### 2.1. Differences in pI Values between High-Abundance Human Plasma Proteins and Snake Venom Proteins

To determine the differences in pI values of proteins abundant in human plasma and venom proteins, we calculated pI values of the top 14 abundant proteins in plasma and venom proteins from *B. multicinctus* and *N. atra* using the online software UniProt Compute pI/Mw. As shown in Appendix A, pI values of the 14 high-abundance proteins in human plasma ranged from 4.6 to 7.2. In contrast, pI values of the major toxic proteins in *B. multicinctus* venom, such as α-bungarotoxin and β-bungarotoxin, were >7.5, ranging from 7.57 to 10.08. Similarly, pI values of most *N. atra* venom proteins were >8. Only a few components belonging to protein families of phospholipase A_2_, snake venom metalloproteinase, and nerve growth factor have lower pI values (4.9 to 7.68). Our results indicate that the pI values of the majority of venom proteins from *B. multicinctus* and *N. atra* are distinct from those of the top 14 plasma proteins, supporting the feasibility of separating major venom components from human plasma proteins based on their charges.

### 2.2. SCX-HPLC for Separating Venom Proteins from Human Plasma Proteins

To ascertain whether *B. multicinctus* and *N. atra* venom proteins could be efficiently separated from plasma proteins based on charge, human plasma and both snake venom samples were analyzed via SCX-HPLC, respectively. According to chromatography patterns, the major peak of the human plasma proteins appeared within 10 min (Figure 1). In analysis of *N. atra* venom, one dominant peak was eluted within 10 min with a retention time almost equal to that of human plasma, while the retention times of the two other peak signals were near 25 and 45 min, respectively (Figure 1A). In analysis of *B. multicinctus* venom, the two most abundant peaks were observed near 15 min and 45 min apart from the peak before 10 min that co-eluted with the major peak of human plasma proteins (Figure 1B). The results collectively suggest that the charges of high-abundance human plasma proteins are distinct from those of several venom proteins from both *B. multicinctus* and *N. atra*, which should allow efficient separation of proteins from snake venom and human plasma during ion exchange chromatography and allow determination of the ion concentration suitable for isolation of venom proteins from plasma matrix.

### 2.3. Development of Methodology for Efficient Enrichment of Snake Venom Proteins from Human Plasma

To determine the suitable ion concentration for separating venom proteins from human plasma, snake venom and plasma proteins were captured using a homemade SCX tip column and eluted stepwise with buffers containing different concentrations of NaCl. Since elution buffers contain high concentrations of salt, each eluted fraction was precipitated with acetone prior to SDS-PAGE. As detected in the gel images, the majority of plasma proteins were not retained with SCX resin and eluted with buffer lacking NaCl (Figure 2). In contrast, most *N. atra* venom proteins were retained in the SCX tip column and eluted as two protein fractions (Figure 3A). The first fraction was eluted with buffers containing 0–40 mM NaCl, and the second with buffers containing 100–300 mM NaCl. When the same experimental conditions were applied to a mixture containing *N. atra* venom and human plasma proteins, the eluted profiles indicated that, as expected, the second fraction of venom proteins eluted with 100–300 mM NaCl was clearly separated from human plasma proteins (Figure 3B).

Upon application of the same experimental design to *B. multicinctus* venom and high-abundance human plasma proteins, similar results were obtained. *B. multicinctus* venom proteins were separated into two groups using the SCX tip column (Figure 4A). The first group of proteins appeared in fractions eluted with 0–60 mM NaCl and the second group with buffer containing 150–250 mM NaCl. Consistent with the above findings, the eluted protein profile clearly showed separation of venom proteins eluted with 150–250 mM NaCl from human plasma proteins (Figure 4B). Overall, these results are in keeping with expectations based on our estimation of pI values of venom proteins from both snakes and abundant proteins from human plasma. Accordingly, buffer containing 80 mM NaCl was utilized as wash buffer for the majority of plasma proteins, and buffer containing 300 mM NaCl was utilized as elution buffer for venom proteins retained in the SCX tip column.

### 2.4. Identification of SCX Tip Column-Enriched Snake Venom Proteins

To determine the identities of enriched venom proteins, marked protein bands on the gel (Figure 3B and Figure 4B) were excised, digested with trypsin, and analyzed via liquid chromatography-tandem mass spectrometry (LC-MS/MS). Among enriched proteins from *N. atra* venom, bands with lower molecular weight (~10 kDa) were identified as cytotoxins (CTX), probable weak neurotoxin, and long neurotoxin homolog, and those with higher molecular weight (~25 kDa) as the cysteine-rich venom protein natrin (Table 1). In addition, enriched protein from *B. multicinctus* venom was identified as beta-bungarotoxin (BBTX) (Table 1). Since this protein has two chains (A and B) linked via a disulfide bond, two protein bands were evident in reducing SDS-PAGE (Figure 4). According to the intensities of protein bands, CTXs and BBTX were determined as the predominantly enriched proteins from *N. atra* and *B. multicinctus* venom types, respectively.

### 2.5. Evaluation of Venom Protein Enrichment Efficiency Via the SCX Tip Column

To evaluate the recovery rates of venom proteins, 5 μg BBTX was subjected to SCX tip column enrichment and enriched samples analyzed via SDS-PAGE with Coomassie blue staining. The recovery rate of BBTX during acetone precipitation, which is usually conducted to rapidly remove high salts from eluted fractions for SDS-PAGE analysis, was additionally evaluated. Compared to the amount of input, recovery rate of acetone precipitation was ~98%, and recovery of SCX tip column enrichment in conjunction with acetone precipitation was ~99% (Figure 5A). While small variations are possible due to operational errors, the results clearly suggest a >95% recovery rate via SCX tip column enrichment for BBTX. In addition, enrichment performance remained consistent between each manually prepared SCX tip column (Appendix A).

To further evaluate the ability to remove background plasma during the enrichment process, human plasma containing 1 mg protein was captured using the SCX tip column and eluted stepwise with buffer containing 0, 80, and 300 mM NaCl (Figure 5B). Each eluted fraction was quantified using the BCA protein assay kit and the proportion of plasma proteins calculated. Our results showed that >95% plasma proteins were depleted upon washing with 0 mM NaCl and only 0.54% remained in the fraction eluted with 300 mM NaCl buffer, clearly indicating that the majority of plasma proteins are efficiently removed with this technique.

### 2.6. Combination of SCX-Tip Column Enrichment with Snakebite Detection Assays

We hypothesized that application of SCX tip column enrichment should facilitate the concentration and purification of venom proteins from clinical samples to aid in the development of efficient bioassays for snakebite detection (Figure 6). To evaluate the feasibility of SCX tip column enrichment, samples prepared by mixing snake venom with human plasma to mimic clinical specimens from snakebite patients were enriched using the SCX tip column and subsequently tested with lateral flow strips developed previously [9] to distinguish between hemorrhagic and neurotoxic snake venoms in Taiwan. A positive signal would appear on the hemorrhagic test line (H line) when the test sample contained *T. stejnegeri* or *P. mucrosquamatus* venom proteins and on the neurotoxic test line (N line) in cases where the test sample contained *B. multicinctus* or *N. atra* venom proteins. The positive signal of the control line signified successful movement of HSS-Ab (hemorrhagic species-specific antibody)- or NSS-Ab (neurotoxic species-specific antibody)-conjugated colloidal gold to the top of the lateral flow strip (i.e., as quality control of the strip). Snake venom was serially diluted with human plasma into four concentrations (50, 10, 5, and 1 ng/mL) and subjected to two different tests. One sample was processed via SCX tip column enrichment before being subjected to the rapid test, and the other was directly examined with the rapid test.

Following SCX tip column enrichment, signals of *B. multicinctus* samples were significantly magnified (Figure 7A). The sample with the lowest concentration (1 ng/mL) still displayed visible bands and detectable signals on the N line compared with the blank control (Figure 7B). The detection limit was enhanced from 5 ng/mL to 1 ng/mL with this process. Consistently, the signal from *N. atra* venom samples was clearer after enrichment (Figure 8A). Following enrichment, the 5 ng/mL sample displayed a visual band on the N line with a relative signal (measured via densitometry analysis) three times higher than that of the blank control and the original sample prior to enrichment. On the other hand, we observed no detectable signal on the N line of the strip with the 1 ng/mL sample before and after SCX tip column enrichment (Figure 8B). These results support our hypothesis that the SCX tip column-based enrichment method developed in this study could successfully increase the sensitivity of lateral flow strip assay for detection of *Elapidae* snake venoms. Furthermore, this device may be further incorporated with other bioassays to improve snakebite diagnosis worldwide.

## 3. Discussion

Here, we examined the hypothesis that different isoelectric points (pI values) of plasma proteins and venom proteins could be used for separation via a SCX tip column-based protocol with buffers containing different NaCl concentrations. Our newly developed enrichment method successfully removed >95% plasma protein background and enhanced the detection limit of the lateral strip assay for snake venom proteins.

A number of recent studies have reported one-step purification methods for venom protein enrichment [11,12,13]. For instance, affinity chromatography-based methods can be effectively used to obtain high-purity (~90%) targeted toxins. However, this procedure is time-consuming, requiring 1–2 h to capture proteins from the matrix [12,13]. Additionally, elution buffers of this methodology usually require extreme pH, necessitating eluted proteins to be processed by buffer exchange or diluted with neutralizing solution before further application. Therefore, the affinity chromatography-based method is appropriate to enrich venom proteins for further pharmacological application or research, but not suitable for combination with bioassays for snake venom detection. Our SCX tip column enrichment method is similar to that of Guan et al. (2018) [11], which is also based on solid-phase extraction (SPE). They used a commercial mixed-mode SPE cartridge to trap cobratoxin, a neurotoxin from *Naja* species, which was eluted with 70% ACN buffer. The high concentration of ACN in elution buffer was applied for further LC-MS/MS analysis. However, high levels of organic solvents disrupt antibody-antigen interactions and may not be appropriate for combination with most antibody-based bioassays and biosensors. The SCX tip column-based method developed in this study requires only 5–10 min for target protein enrichment, which is suitable for the immediate diagnostic requirement of snakebite management in an emergency. Moreover, the buffer system used in this enrichment technique does not require extreme pH or high concentrations of organic solvents. The NaCl-based buffer can preserve affinity interactions between antibodies and antigens and is thus more applicable while combining with antibody-based assay platforms. In addition to lateral flow strip assay, SCX tip column enrichment may be efficiently combined with other bioassays or biosensors for snake venom detection, such as ELISA [14], surface plasmon resonance [15,16], and optical biosensors [17], to achieve the goal of early detection with high sensitivity.

In the future, this method can be applied to enrich proteins from other medically significant venom samples to facilitate accurate snakebite diagnosis. According to our preliminary survey, several venom proteins other than *N. atra* and *B. multicinctus* from *Elapidae* snakes have the property of high pI value and may thus be efficiently separated from human plasma via SCX tip column enrichment. For example, *Ophiophagus hannah* venom is mainly composed of 3FTX with a pI value of around 8.5 [18] and *Dendroaspis polylepis* venom mainly comprises Kunitz-type serine protease inhibitors with a pI value of 9.93 [19]. Both snake venoms are potential candidates for application of our enrichment method. However, this technique has a number of limitations. Our study design is based on the difference in pI values between venom and plasma proteins, and its application may not be appropriate to enrich venom samples whose major protein components have lower pI values than plasma proteins. For instance, the venom of *Bothrops atrox*, a viper from Brazil, is mostly composed of snake venom metalloproteinases with a pI value of around 5.8, which is similar to that of high-abundance human plasma proteins [20], and would therefore not be suitable for protein enrichment using the SCX tip column.

## 4. Conclusions

In conclusion, we have developed an SCX tip column enrichment method that can be used to rapidly enrich snake venom proteins from human plasma. This technique facilitates removal of >95% of the plasma background, thereby enhancing the venom protein signal on lateral flow strip assays, leading to an increase in the detection limit from 5 ng/mL to 1 ng/mL for *B. multicinctus* venom and 10 ng/mL to 5 ng/mL for *N. atra* venom. Effective concentration of target venom proteins and removal of the background matrix from biological samples by the SCX tip column not only improves the sensitivity of the lateral flow strip assay, but also has the potential to enhance the performance of other snake venom detection assays or biosensors. In addition, target venoms are not limited to Taiwanese cobra and krait. Other *Elapidae* snake venoms containing proteins with higher pI values are expected to be enriched using this easily operable device. Our novel rapid, simple, and cost-effective method should aid in improving the clinical utility of snakebite diagnostic assays.

## 5. Materials and Methods

### 5.1. Snake Venom

Lyophilized venom of *N. atra* and *B. multicinctus* was obtained from the Centers for Disease Control, R.O.C. (Taiwan). Purified beta-bungarotoxin was purchased from Biotium (Hayward, CA, USA).

### 5.2. Strong Cation Exchange High-Performance Liquid Chromatography (SCX-HPLC)

The sample was dissolved in solution A (50 mM Na_3_PO_4_, pH 6.8) and separated via HPLC using a Mono S 4.6/100PE column. The flow rate was set to 1 mL/min and the column eluted with a linear gradient of solution A and solution B (50 mM Na_3_PO_4_, 1 M NaCl, pH 6.8) as follows: Isocratic (0% solution B for 3 min), followed by linear gradients of 0−70% solution B for 40 min, 70–100% solution B for 2 min, isocratic 100% B for 3 min, linear gradient of 70–0% solution B for 2 min, and subsequent re-equilibration with 0% solution B for 10 min. Peaks were detected by monitoring absorbance at 214 nm.

### 5.3. Separation of Venom and Plasma Proteins Using the Strong Cation Exchange (SCX) Tip Column

The tip column composed of Axygen^®^ 200 µL universal fit pipet tips (Product Number: TR-222-Y, Corning, Arizona, USA) and a glass fiber filter (Product Number: 240-1, Cambridge Technology Inc., Massachusetts, U.S.A) was manually prepared. The detailed process of generating the tip column is presented in Appendix A. Each tip column was packed with 20 μL.

SOURCE 30S medium (GE healthcare, IL, USA), washed with 100 μL elution buffer (300 mM NaCl, 10 mM Na_3_PO_4_, pH 7.4, 25% ACN) and equilibrated with 100 μL equilibration buffer (10 mM Na_3_PO_4_, pH 7.4, 25% ACN). Two volumes of wash buffer (80 mM NaCl, 10 mM Na_3_PO_4_, pH 7.4, 25 % ACN) were added to the sample (200 µl buffer to 100 µL sample) and the diluted sample flushed through the SCX tip column twice. The column was subsequently washed with 100 μL of wash buffer three times. Finally, enriched proteins were eluted with 60 μL buffer. Enriched samples were used immediately or stored at −20 °C for further use.

### 5.4. Acetone Precipitation

Protein sample (200 µL) was mixed with 800 μL of 100% acetone and incubated at −20 °C overnight to precipitate proteins. Samples were centrifuged at 16,000× *g* for 10 min at 4 °C. The resulting supernatant was removed and precipitated sample dried using SpeedVac. Dried samples were stored at −20 °C or dissolved in sample buffer for SDS-PAGE.

### 5.5. In-Gel Tryptic Digestion

Selected protein bands were excised from the gel and subjected to in-gel tryptic digestion, as described by Lin et al. [21]. Gel pieces were destained with 40% ACN containing 30 mM ammonium bicarbonate for 15 min. After removal of the solution, the gel was incubated in 25 mM ammonium bicarbonate for 15 min, which was removed, followed by incubation in 100% ACN for 5 min. Following removal of ACN, the gel was reduced with 10 mM dithiothreitol at 56 °C for 45 min. Dithiothreitol was washed off, and the gel alkylated with 55 mM iodoacetamide at room temperature in the dark for 30 min. After washing off iodoacetamide, the gel was incubated in 40% ACN containing 30 mM ammonium bicarbonate for 15 min and finally, 100% ACN for 5 min. ACN was washed off, proteins in the processed gel pieces digested with freshly prepared 20 μg/mL trypsin solution (Promega, Madison, WI, USA) in 25 mM ammonium bicarbonate at 37 °C for 16 h, and extracted with 100% ACN containing 1% TFA. Tryptic peptide extracts were concentrated with SpeedVac and stored at −20 °C before use.

### 5.6. LC-MS/MS Analysis

Each peptide sample was reconstituted with 0.1% formic acid (FA) and analyzed on a nano-LC–LTQ-Orbitrap Hybrid Mass Spectrometer (Thermo Fisher, CA, USA) as described previously [22]. Briefly, samples were loaded across a trap column (Zorbax 300SB-C18, 0.3 × 5 mm; Agilent Technologies, Wilmington, DE, USA) at a flow rate of 0.2 μL/min in HPLC buffer (0.1% FA) and separated on a resolving 10 cm analytical C18 column (inner diameter, 75 μm) using a 15 μm tip (New Objective, Woburn, MA, USA). Peptides were eluted using a linear gradient of 0–10% HPLC elution buffer (99.9% ACN containing 0.1% FA) for 3 min, 10–30% buffer B for 35 min, 30–35% buffer B for 4 min, 35–50% buffer B for 1 min, 50–95% buffer B for 1 min, and 95% buffer B for 8 min, with a flow rate of 0.25 μL/min across the analytical column. The resolution of Orbitrap was 30,000, and the ion signal of (Si(CH3)2O)6H+ at 445.120025 (m/z) used as a lock mass for internal calibration. One MS scan alternating with six MS/MS scans for the 10 most abundant precursor ions was applied. The m/z values selected for MS/MS were dynamically excluded for 180 s. For MS scans, the m/z value of the scan range was 400–2000 Da. For MS/MS scans, >1 × 10^4^ ions accumulated in the ion trap to generate spectra. MS and MS/MS spectra were acquired using one scan with maximum fill times of 1000 and 100 ms, respectively.

### 5.7. Database Searches and Bioinformatics Analysis

Raw MS data files were analyzed using Proteome Discoverer Software (version 1.3.0.339; Thermo Fisher, San Jose, CA, USA) and searched against the UniProt database (Taxonomy: Other lobe-finned fish and tetrapods) using the MASCOT search engine (version 2.2; Matrix Science, London, UK). The enzyme specificity parameter was set to “trypsin” and one missed cleavage was allowed. Carbamidomethylation of cysteines was set as a static modification, and oxidation of methionine, acetyl (protein N-term), and Gln- > pyro-Glu (N-term Q) set as dynamic modifications. The tolerance of MS was 10 ppm, and that of MS/MS was 0.5 Da. The decoy database search approach was assessed for peptide identification and threshold of target false discovery rate (FDR) estimated as <0.01. Each reported protein ID should have at least two peptides present in the sample, at least one of which is the unique peptide for the reported protein.

### 5.8. Venom Detection with Lateral Flow Strips

Lateral flow strips for detection of *N. atra* and *B. multicinctus* venom proteins were designed as described previously [9]. Briefly, conjugate pads were saturated with colloidal gold conjugated with HSS-Ab (hemorrhagic species-specific antibodies) or NSS-Ab (neurotoxic species-specific antibodies) and dried at 37 °C for 1 h before assembly. Nitrocellulose membrane was pasted onto the cardboard followed by conjugated and absorbent pads that overlapped each side of the nitrocellulose membrane by about 2 mm. The sample pad was also laid over the absorbent pad and pasted onto the cardboard. The AGISMART RP-1000 immuno-strip printer (REGA Biotechnology Inc., Taipei, Taiwan) was used to dispense HSS-Abs and NSS-Abs (2 mg/mL) onto hemorrhagic and neurotoxic test lines (H and N lines), respectively, and goat anti-horse IgG antibody (2 mg/mL) (REGA Biotechnology Inc.) onto the control line on the nitrocellulose membrane. The distance between each line was 5 mm. Strips were prepared and assembled in a low-humidity environment, packaged into an aluminum pouch, and stored at room temperature before use. Mimic snake venom samples were subjected to SCX tip column enrichment and the enriched sample (100–200 μL) subsequently diluted with an equal volume of reaction buffer (100 mM borax, 250 nM polyvinylpyrrolidone (PVP)−40 and 1% Triton X-100) in a microcentrifuge tube. Lateral flow strips were directly soaked in the samples, and the results were recorded after a 15-min reaction. The signal intensity on the neurotoxic test line was quantified via densitometry using ImageJ software [23].

## Figures and Tables

**Figure 1 toxins-13-00140-f001:**
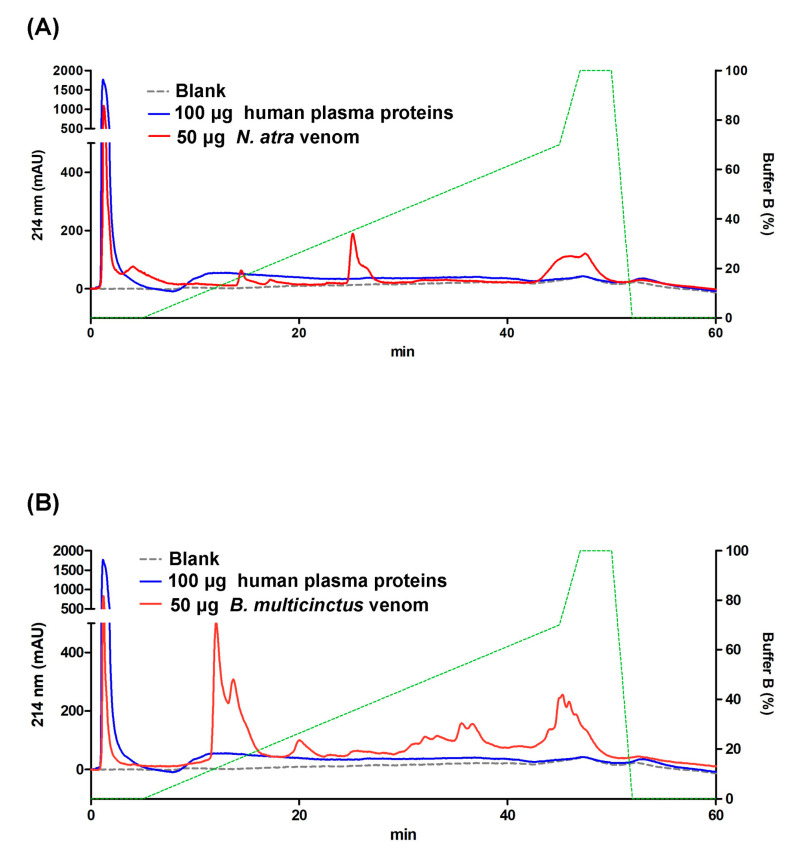
Chromatography patterns of human plasma proteins and *Naja atra* and *B. multicinctus* venom proteins analyzed via strong cation exchange–high-performance liquid chromatography (SCX-HPLC). Human plasma proteins (100 μg), *Naja atra* venom proteins (50 μg), and *B. multicinctus* venom proteins (50 μg) were individually analyzed using SCX-HPLC. The overlapping patterns of human plasma proteins with *Naja atra* and *B. multicinctus* venom proteins are shown in (**A**,**B**), respectively.

**Figure 2 toxins-13-00140-f002:**
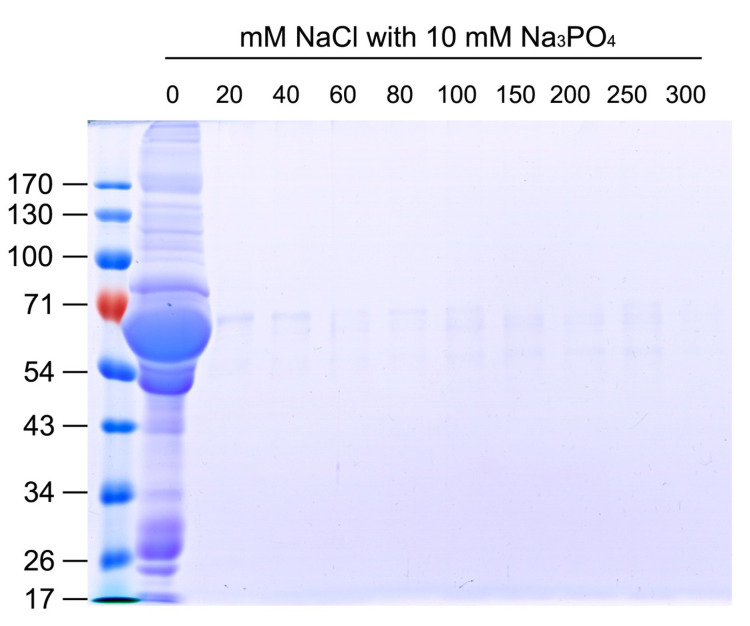
Separation of human plasma proteins with SCX tip column. Human plasma proteins (200 μg) were loaded onto the SCX tip column and eluted with buffer containing different NaCl concentrations. Each eluted fraction was precipitated with acetone and analyzed via 10% SDS-PAGE. Protein bands were visualized using Coomassie blue staining.

**Figure 3 toxins-13-00140-f003:**
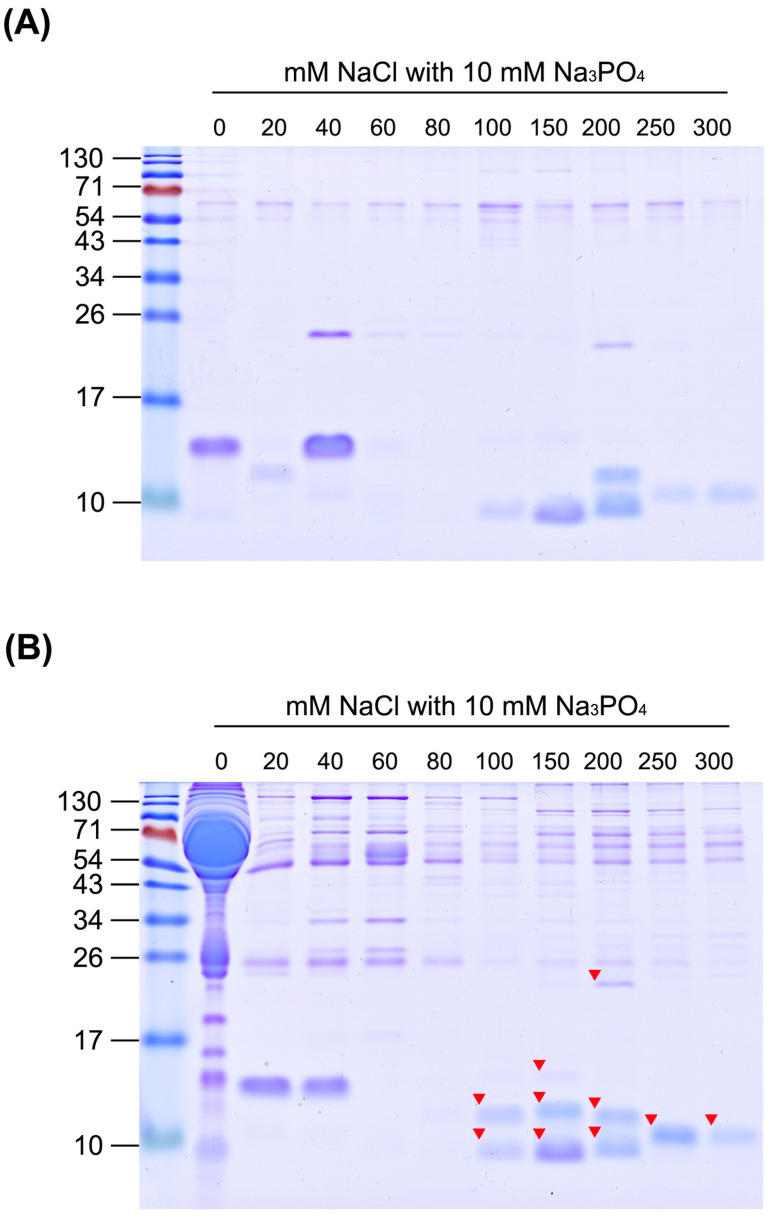
Separation of *N. atra* venom proteins from human plasma proteins with SCX tip column. (**A**) *N. atra* venom proteins (25 μg) and (**B**) a mixture containing *N. atra* venom proteins (25 μg) and human plasma proteins (100 μg) were, respectively, loaded onto SCX tip columns and stepwise-eluted with buffers containing different concentrations of NaCl. All eluted fractions were precipitated with acetone and half the precipitate analyzed using 15% SDS-PAGE. Protein bands were visualized via Coomassie blue staining. Arrowheads denote the bands excised for protein identification using liquid chromatography-tandem mass spectrometry (LC-MS/MS).

**Figure 4 toxins-13-00140-f004:**
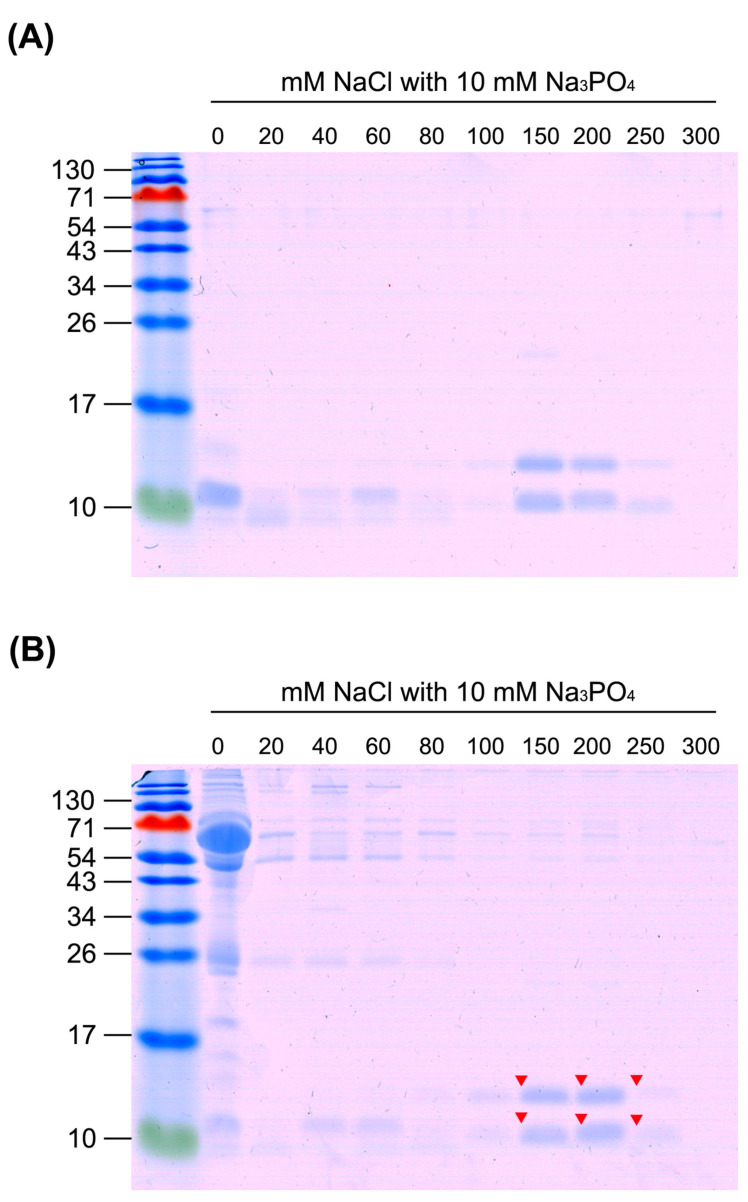
Separation of *B. multicinctus* venom proteins from human plasma proteins with the SCX tip column. (**A**) *B. multicinctus* venom proteins (25 μg) and (**B**) a mixture of *B. multicinctus* venom proteins (25 μg) and human plasma proteins (100 μg) were, respectively, loaded onto SCX tip column and stepwise-eluted with buffers containing different concentrations of NaCl. All eluted fractions were precipitated with acetone and half of the precipitate analyzed via 15% SDS-PAGE. Protein bands were visualized with Coomassie blue staining. Arrowheads denote the protein bands excised for protein identification using LC-MS/MS.

**Figure 5 toxins-13-00140-f005:**
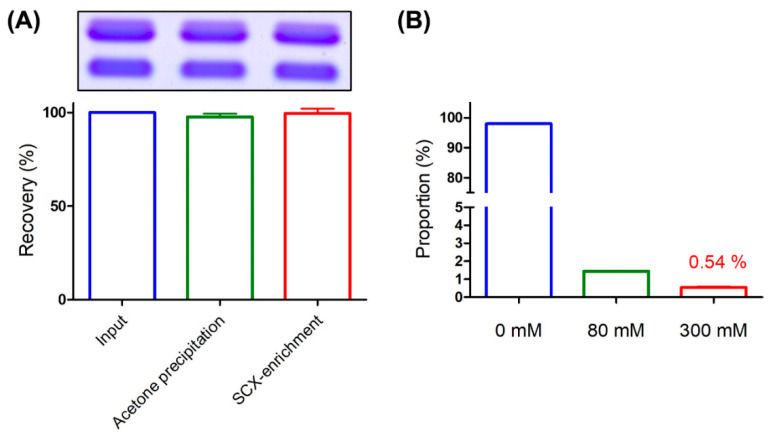
Evaluation of venom protein enrichment efficiency with the SCX tip column. (**A**) Purified BBTX (5 μg) was subjected to acetone precipitation and SCX tip column enrichment in conjunction with acetone precipitation. SDS-PAGE with Coomassie blue staining was conducted to evaluate the recovery rate of BBTX by comparing with the amount of input. (**B**) Human plasma (1 mg protein) was captured with a SCX tip column and eluted with 0, 80, and 300 mM NaCl buffer. Each eluted fraction was quantified using BCA and the proportion of plasma proteins in individual fractions calculated. The bar charts depict the average of triplicate results with standard deviation (SD).

**Figure 6 toxins-13-00140-f006:**
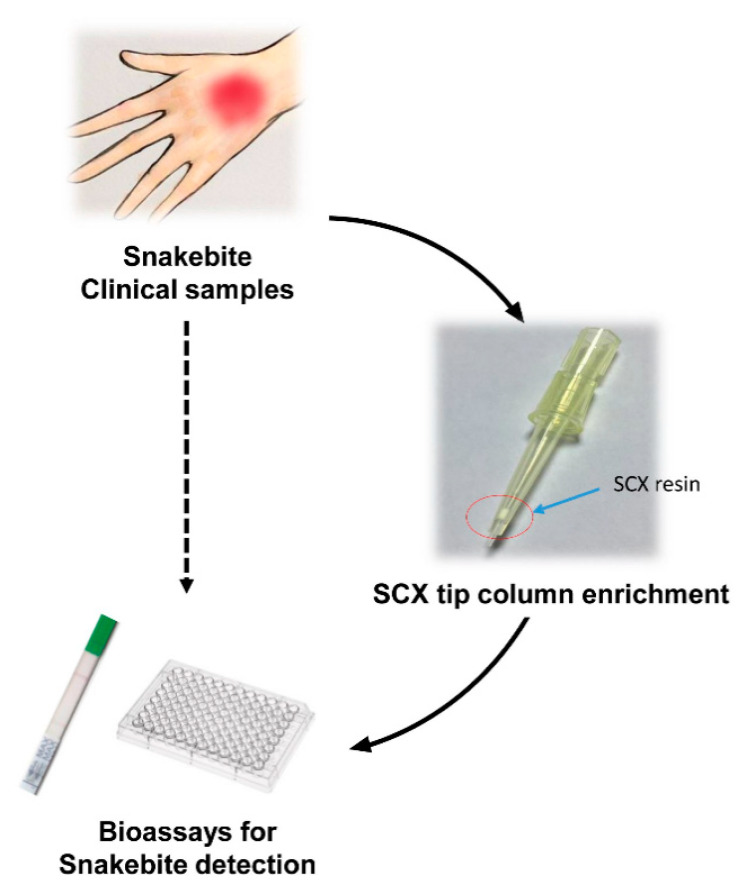
Flowchart of hypothetical usage of SCX tip column-based enrichment of venom proteins.

**Figure 7 toxins-13-00140-f007:**
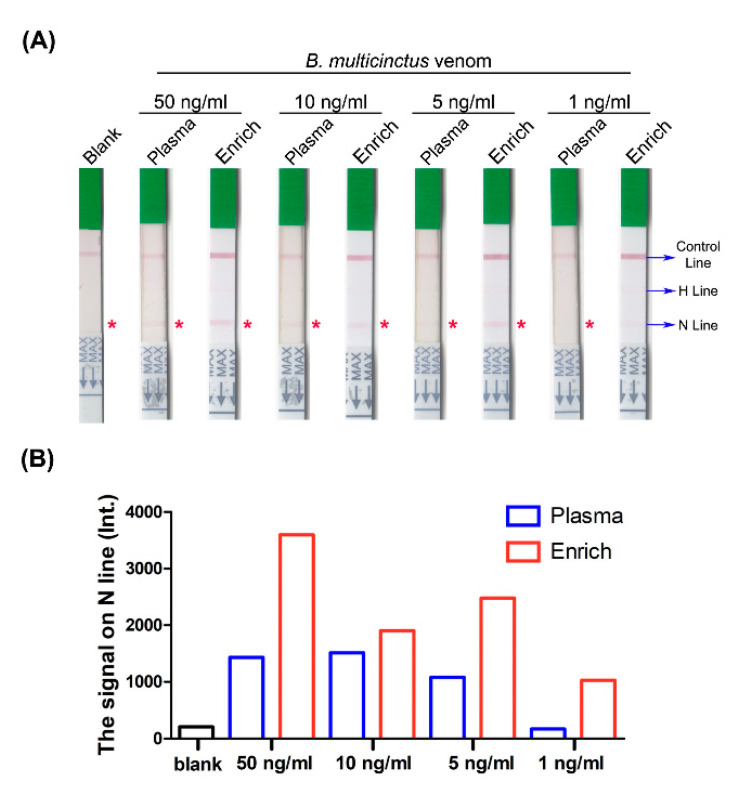
Detection of *B. multicinctus* venom proteins in human plasma before and after SCX tip column enrichment via the lateral flow strip assay. (**A**) *B. multicinctus* venom was serially diluted with human plasma into four concentrations, processed with or without SCX tip column enrichment, and tested with lateral flow strips. Blank, plasma sample without venom; Plasma, venom-containing plasma before enrichment; Enrich, venom-containing plasma after enrichment; H line, hemorrhagic test line; N line, neurotoxic test line. The asterisk (*) denotes the position of N line on each strip. (**B**) Quantification of the signal on the N line via densitometry analysis using ImageJ software.

**Figure 8 toxins-13-00140-f008:**
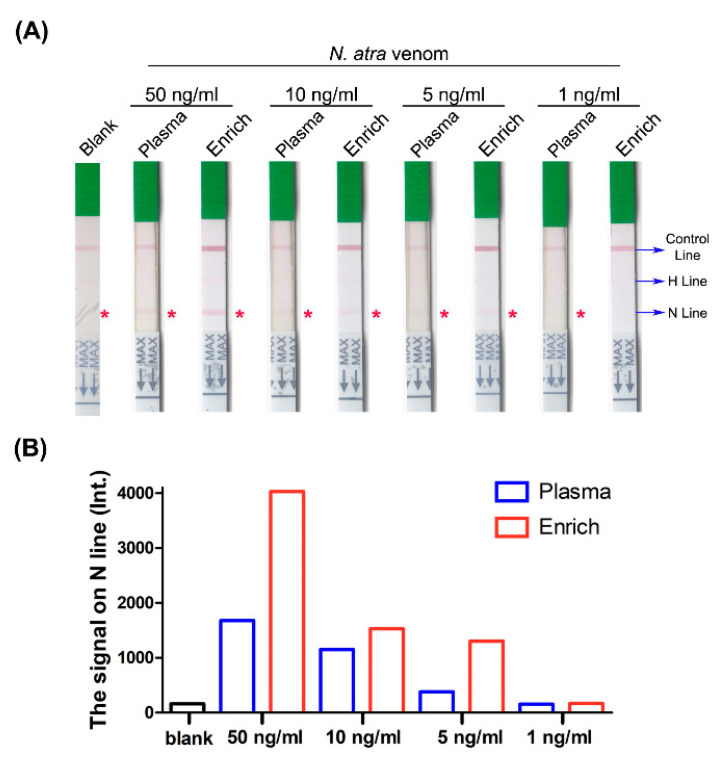
Detection of *N. atra* venom proteins in human plasma before and after SCX tip column enrichment with the lateral flow strip assay. (**A**) *N. atra* venom was serially diluted with human plasma into four concentrations, processed with or without SCX tip column enrichment, and tested with lateral flow strips. Blank, plasma sample without venom; Plasma, venom-containing plasma before enrichment; Enrich, venom-containing plasma after enrichment; H line, hemorrhagic test line; N line, neurotoxic test line. The asterisk (*) denotes the position of N line on each strip. (**B**) Quantification of the signal on the N line via densitometry analysis using ImageJ software.

**Table 1 toxins-13-00140-t001:** Summary of proteins identified in SCX-enriched venom protein samples via LC-MS/MS.

Species	Sample	Accession	Description	Coverage (%) ^1^	Peptides ^2^	PSMs ^3^	MW [kDa]	pI
***N. atra***	100mM-1	P60304	Cytotoxin 1	35.8	4	10	9.0	8.7
	100mM-2	P60304	Cytotoxin 1	71.6	11	229	9.0	8.7
		P80245	Cytotoxin 6	58.0	8	103	9.0	8.9
		P01442	Cytotoxin 2	51.9	7	86	9.0	8.9
		P01443	Cytotoxin 4	51.9	7	86	9.1	9.0
	150 mM-1	P60304	Cytotoxin 1	35.8	5	12	9.0	8.7
		P80245	Cytotoxin 6	43.2	5	10	9.0	8.9
	150 mM-2	O93422	Long neurotoxin homolog	22.1	2	6	9.8	8.7
		Q9YGI2	Probable weak neurotoxin NNAM1	22.1	2	6	9.8	8.5
		Q9YGI4	Probable weak neurotoxin NNAM2	25.6	2	10	9.9	8.6
	150 mM-3	P60304	Cytotoxin 1	71.6	11	356	9.0	8.7
		P80245	Cytotoxin 6	71.6	12	164	9.0	8.9
		P01442	Cytotoxin 2	59.3	9	124	9.0	8.9
		P01443	Cytotoxin 4	59.3	9	124	9.1	9.0
	200 mM-1	Q7ZZN8	Cysteine-rich venom protein natrin-2	33.6	9	63	26.2	8.7
	200 mM-2	Q9YGI2	Probable weak neurotoxin NNAM1	36.1	4	99	9.8	8.5
		O93422	Long neurotoxin homolog	36.1	4	99	9.8	8.7
		Q9YGI4	Probable weak neurotoxin NNAM2	25.6	2	37	9.9	8.6
	200 mM-3	P60301	Cytotoxin 3	60.5	9	178	9.0	9.0
		P60304	Cytotoxin 1	65.4	9	161	9.0	8.7
		P80245	Cytotoxin 6	58.0	9	115	9.0	8.9
	250 mM-1	P62375	Cytotoxin A5	45.8	4	44	9.3	9.0
	300 mM-1	P62375	Cytotoxin A5	46.0	6	17	9.3	9.0
***B. multicinctus***	150 mM-1	P00617	Basic phospholipase A2 beta-bungarotoxin A1 chain	21.1	2	5	16.2	7.5
	150 mM-2	P00987	Kunitz-type serine protease inhibitor homolog beta-bungarotoxin B1 chain	28.2	3	17	9.6	8.7
	200 mM-1	P00617	Basic phospholipase A2 beta-bungarotoxin A1 chain	61.9	9	44	16.2	7.5
	200 mM-2	P00987	Kunitz-type serine protease inhibitor homolog beta-bungarotoxin B1 chain	48.2	5	26	9.6	8.7
	250 mM-1	P00617	Basic phospholipase A2 beta-bungarotoxin A1 chain	53.1	7	17	16.2	7.5
	250 mM-2	P00987	Kunitz-type serine protease inhibitor homolog beta-bungarotoxin B1 chain	30.6	3	9	9.6	8.7

^1^ Coverage (%): Displays by default the percentage of the protein sequence covered by identified peptides. ^2^ Peptides: Displays the number of distinct peptide sequences in the protein. ^3^ PSMs: Displays the total number of identified peptide sequences (peptide spectral matches) for the protein, including those redundantly identified.

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
