# Peer review of "Rapid and Efficient Enrichment of Snake Venoms from Human Plasma Using a Strong Cation Exchange Tip Column to Improve Snakebite Diagnosis"

_toxins, 2021, doi:10.3390/toxins13020140_

Round 1

Reviewer 1 Report

The authors have conducted a study that is investigating methods to increase the concentration of snake venom in a blood sample taken from a snakebite victim. The study successfully uses an SCX-tip column to remove plasma proteins from a plasma/venom sample, thereby increasing increasing the proportion of venom proteins in the final sample. This increases the sensitivity of the rapid diagnostic kit subsequently used to identify the species of snake.

This is an important area of research and shows potential as a useful method aiding snakebite treatment. Congratulations.

However, there are aspects of the study that require improvement before it can be published.

  1. The grammar and writing style needs to be improved. Much of it is understandable, however there are several paragraphs throughout the manuscript that are unclear. I understand that it is very difficult for authors who do not speak English as their first language (this is unfortunately an unfair disadvantage for you). I recommend enlisting someone who speaks english fluently or hiring an editor.
  2. Some brief background information on the science behind the rapid diagnostic test and how the strips actually work to detect venoms would be useful. This will give the your new method greater context and significance.
  3. Lines 48 & 49. Antivenoms are only administered if truly required due to the adverse risks associated with antivenoms. Many envenoming events do not require antivenom if the concentration of venom is low. Are venom concentrations below 5-10 ng/mL clinically significant? If they are clinically significant, would antivenoms be administered at these low concentrations or would symptoms and sequelae be monitored and treated instead? If venom concentrations below 5-10 ng/mL do not require antivenom then the significance of this study is limited. This issue should be addressed in the manuscript.
  4. Line 70-72. What is are the implications for this - are there any? If several SVMPs and PLA2s have pI values within the same range as plasma proteins, does this mean that the assay is going to be of limited use for many snake species whose venoms are mostly composed of SVMP or PLA2? 
  5. The context and significance of your results could be more clearly stated.
  6. Figure 7 - What do the control line, H line and N line represent? It is hard to see the lines properly based on the resolution of this figure - i can’t tell if I can see a line or not in some cases. Also - it would be good to include a control strip that has plasma and no venom so that we can compare the lower concentrations to a blank.
  7. The broader application of the assay should be more clearly stated. What are its limitations? Are these assays going to be useful for many other medically significant venoms?
  8. Line 282 - diluted to what concentration?

The paper shows promise to become an interesting and useful publication once these issues are addressed. 

Reviewer 2 Report

Brief summary

The authors describe a strong cation exchange (SCX), pipette tip–based, column chromatography protocol (SCX-tip column) to rapidly enrich venom proteins from human plasma. This application is based on differences between venom and plasma protein charges. The entire process of venom enrichment can be completed in 10-15 minutes. The authors demonstrate how the simple method may be useful to improve the clinical utility of snakebite diagnostic assays. 

Broad comments

The manuscript is, in general, well written. Issues of English syntax, however, may need to be addressed. Examples are included in the specific comments.  The manuscript does not indicate if the experiments were reproducible. 

The methods described in this manuscript could be useful for clinical evaluation of envenomation if tip-column production was standardized and chromatographic results validated. Validation of SCX tip-column production and results would be needed prior to acceptance for publication, in my opinion.

Specific comments

The term “rapid test” used in the manuscript was not clearly defined.  Consider adding the term, parenthetically, after “lateral flow strip assays” on Line 16 in the abstract, i.e. “lateral flow strip assays (rapid test)”.

English syntax, examples:

Line 48 - “… limit of the rapid test…”

Line 70 - “Only a few… “

Sentence on Lines 56-57 - Something seem to be missing at the end of the sentence.

Line 80 – The term “spectrum” should be replaced by “pattern” (to be consistent with the title of Figure 1).

Methods:

Section 4.3 Separation of venom and plasma proteins by strong cation exchange (SCX) tip column

More detail about how to make tip-columns would be helpful, including materials used (tips, pipettor, etc.).

How many tip-columns were used? How many times can one tip-column be used? Is there evidence for reproducible results from the same column, and/or reproducible results using different columns?

Reviewer 3 Report

The authors describe experiments to simply enrich two snake venoms mixed with human plasma proteins. Plasma proteins: was that a commercial preparation or just human plasma from a blood sample? In the latter case, was that standardized? The authors use rather high concentrations of venom from both species, i.e. 25 microgram venom and 100 microgram plasma proteins, but the results, as shown in Fig. 3 and 4, i.e. the bands claimed to be specific for venom proteins are rather faint. The same applies to the demonstration of the bands in the lateral flow strip assays. Except in case of levels of 50 ng venom/ml, lines below these concentrations are hardly detectable (what  represents H- and N-line? Explain). From my view, sensitivity of the assays is not really increased after the enrichment method and I dont think that this test system is hardly applicable, when it needs to be performed in practice, e.g. in case of snakebite. Why not testing the method using blood from a patient bitten by one of the two snakes?  

Round 2

Reviewer 1 Report

The paper has been significantly improved, congratulations to the authors. I have only three minor comments for the revised edition. 

1 - In your responses to my first comments, you have informed me that the standard practice in Taiwan is to give antivenom to all snakebite patients regardless of venom serum levels. This is not common practice globally as many nations do not give antivenom unless absolutely necessary, and will avoid it if symptoms can be clinically managed. I therefore strongly suggest that you include this information (i.e. that antivenom is always given for envenoming in Taiwan) in your introduction as it substantially increases the significance of your study.

2 - lines 44 and 51, there is a space immediately after the reference ([5] and [8-10]) and before the comma that should be deleted.

3 - In line 302, I think the sentence is missing a word: 

"concentration of target venom proteins and removal the background matrix" should possibly say "concentration of target venom proteins and removal OF the background matrix".

Reviewer 2 Report

The authors have improved the manuscript and adequately addressed concerns of the reviewer.

Author Response

We thank the reviewer's efforts in reviewing our paper.

Reviewer 3 Report

no further comments

Author Response

(The authors gave the same response as above.)
